# Who Does Your Algorithm Fail? Investigating Age and Ethnic Bias in the MAMA-MIA Dataset

## Abstract

Deep learning models aim to improve diagnostic workflows, but fairness evaluation remains underexplored beyond classification, e.g. in image segmentation. Unaddressed segmentation bias can lead to disparities in the quality of care for certain populations, potentially compounded across clinical decision points and amplified through iterative model development. Here, we audit the fairness of the automated segmentation labels provided in the breast cancer tumor segmentation dataset MAMA-MIA. We evaluate automated segmentation quality across age, ethnicity, and data source. Our analysis reveals an intrinsic age-related bias against younger patients that continues to persist even after controlling for confounding factors, such as data source. We hypothesize that this bias may be linked to physiological factors, a known challenge for both radiologists and automated systems. Finally, we show how aggregating data from multiple data sources influences site-specific ethnic biases, underscoring the necessity of investigating data at a granular level.

## 1 Introduction

Automated segmentation of breast tumors is a critical step in diagnosis, monitoring, treatment planning, and advancing robot-assisted surgeries [Michael et al., 2021, Benjelloun et al., 2018]. Inaccurate segmentation can lead to missed diagnoses, suboptimal treatment plans, and heterogeneous health outcomes across the patient population [Veta et al., 2014]. While recent advancements in deep learning have reached state-of-the-art performance [Isensee et al., 2021], their "fairness"– the principle that a model should not systematically disadvantage certain patient subgroups– remains a critical but often-overlooked aspect in medical image segmentation studies [Larrazabal et al., 2020].

The lack of datasets pairing high-quality imaging with demographic information and clinical metadata has limited the study of bias in medical image segmentation [Suresh and Guttag, 2019]. The MAMA-MIA dataset [Garrucho et al., 2024] addresses this gap, providing a multi-center cohort including detailed demographic information, clinical characteristics, and technical specifications. We perform a careful audit of the automated deep learning segmentations released as part of MAMA-MIA, following the *fairness under unawareness* paradigm [Barocas et al., 2023, Puyol-Antón et al., 2021]. We focus on ethnicity- and age-related disparities, motivated by clinical literature suggesting a correlation between younger patients, breast tissue density, and model performance on tumor detection tasks, caused by challenging tumor delineation for both radiologists and automated systems [Freer, 2015, Kontos et al., 2019, Tiryaki and Kaplanoğlu, 2022].

Our study provides: (i) to our knowledge, a first comprehensive fairness audit examining the intersection of multiple sensitive attributes in the MAMA-MIA breast tumor segmentation dataset, revealing statistically significant age- and ethnicity-based performance disparities; (ii) evidence that aggregating multi-center data and interaction between multiple factors can obscure site-specific ethnic bias; and (iii) a preliminary analysis on whether the bias is caused by lack of representation.

Submitted to 39th Conference on Neural Information Processing Systems (NeurIPS 2025). Do not distribute.

## 2 Dataset and Methodology

**The MAMA-MIA Dataset** is a large, multi-center breast cancer benchmark of dynamic contrast-enhanced magnetic resonance images (DCE-MRI) [Garrucho et al., 2024]. Integrating four cohorts hosted on TCIA[1], it contains 1,507 T1-weighted DCE-MRI cases of female breast cancer patients.

We analyze key demographic and technical attributes with complete data coverage and established clinical relevance to segmentation performance: **Ethnicity:** Caucasian (74.9%), African-American (16.0%), Asian (5.7%), and other minority groups (3.4%); **Age Groups:** Young (<40 years, 23.2%), Middle (40-55 years, 50.1%), Older (>55 years, 26.6%). We discretize the continuous age variable into three bins informed by clinical literature on breast density changes and menopausal status transitions [Boyd et al., 2007, Checka et al., 2012]. The dataset uniquely includes dual annotations: Tumor regions manually segmented by a panel of 16 expert radiologists, serving as ground-truth annotations (gold labels) and automated segmentation masks from a model trained on external data (silver labels). The silver labels come with dual-expert qualitative ratings (Good, Acceptable, Poor, or Missed) assessing their visual quality. This dual annotation structure enables investigation of both model performance disparities and potential label quality biases across subgroups.

A **Fairness Auditing Framework** based on *fairness under unawareness* is adopted, following Chen et al. [2019]. Notably, [Puyol-Antón et al., 2021] conducted a pioneering work auditing deep learning models for cardiac image segmentation, assessing bias in models trained without explicit knowledge of sensitive attributes. We evaluate bias in automatic segmentation quality by comparing silver labels against gold labels using the Dice Score, 95th percentile Hausdorff Distance (HD95), and expert quality ratings. To formally quantify disparities, we measure the Demographic Parity Difference (DPD) $= |P(\hat{y} = 1|A = a) - P(\hat{y} = 1|A = b)|$ and Disparate Impact Ratio (DIR) $= \frac{\min(P(\hat{y}=1|A=a), P(\hat{y}=1|A=b))}{\max(P(\hat{y}=1|A=a), P(\hat{y}=1|A=b))}$. Here, $\hat{y} = 1$ is the beneficial outcome (e.g., a high-performance segmentation), $A$ is the sensitive attribute (e.g., age or ethnicity), and $a, b$ are distinct subgroups within that attribute [Caton and Haas, 2024, Castelnovo et al., 2022].

For these metrics, samples scoring in the top 25% for each metric were classified as high performers. Fairness gap (§) is the absolute difference in mean performance between the highest- and lowest-performing demographic subgroups [Tran and Woo, 2025].

To isolate the effect of representational imbalance, we conducted a controlled experiment using a setup designed to be representative of the original automated model. We trained a standard nnU-Net model using a 5-fold cross-validation scheme on an age-balanced cohort (n=1,047). This cohort was created by downsampling the 'Middle' and 'Older' groups to match the 'Young' group (n=349) and used only expert-annotated gold labels for training.

**Statistical Analysis** was used to assess performance differences across demographic subgroups. We first employ Ordinary Least Squares (OLS) regression [Zdaniuk, 2014] to model the relationship between sensitive attributes and performance metrics. Given that the performance metric distributions were non-normal (confirmed by Shapiro-Wilk tests), we used the non-parametric Kruskal-Wallis H-test to identify significant differences in performance across age and ethnicity groups. Where a significant overall difference was found, we conducted post-hoc pairwise comparisons to identify which specific subgroups differed, using Bonferroni correction. For the analysis of categorical expert ratings, the Chi-square test was used.

## 3 Results and Discussion

**Age-Related Performance Disparities:** Our analysis reveals that the automated silver labels have lower quality for younger patients, a statistically significant disparity that persists beyond simple representational imbalance. Across the complete cohort, segmentation quality improves with age. A baseline OLS regression (`Performance ~ Age`) demonstrates a significant, although small, relation between age and segmentation performance (Dice score: $R^2 = 0.0104$, $p = 0.0001$; HD95: $R^2 = 0.0093$, $p = 0.0009$), as visualized in Fig.1 (Right). These quantitative results are reflected in fairness metrics, which show a notable performance gap; for the Dice score, the DPD was 0.0887, with the 'young' group achieving a high performance at only 70% the rate of the 'older' group (DIR = 0.699).

---

[1]The Cancer Imaging Archive (TCIA) hosts de-identified cancer imaging datasets. Cohorts include: DUKE, I-SPY1 & 2, and NACT.

To determine if this bias was merely a confounding effect of the data source, we adjusted our OLS model to account for the source dataset, fitting a model of the form `Performance ~ AgeGroup + DataSource`. An ANOVA comparison between the baseline and the source-adjusted models confirmed that the source dataset is a significant factor (Dice: $F = 11.76$, $p = 1.3 \times 10^{-7}$; HD95: $F = 11.07$, $p = 3.4 \times 10^{-7}$). Even after this adjustment, the age effect remained highly significant, indicating the bias is not solely attributable to dataset-specific characteristics. This points towards an **intrinsic bias**. Further analysis revealed an interaction effect between age and dataset (Dice: $p = 1.6 \times 10^{-8}$), suggesting the magnitude of age-related bias varies depending on the data source.

**Our controlled experiment on the age-balanced cohort** confirmed the age-related bias is **intrinsic**, with a statistically significant fairness gap of 0.0399 (ANOVA p=0.0260) persisting even after eliminating representational imbalance, as shown in the comparative results in Table1.

Table 1: Age-Stratified Performance

| Age | Balanced Cohort | Automated |
|---|---|---|
| Young | $0.7304 \pm 0.2333$ | $0.8082 \pm 0.2193$ |
| Middle | $0.7333 \pm 0.2253$ | $0.8204 \pm 0.2139$ |
| Older | $0.7703 \pm 0.1899$ | $0.8612 \pm 0.1679$ |
| § | **0.0399** | **0.0530** |
| **p-value** | **0.0260** | **0.0006** |

Figure 1: **Analysis of Age-Related Bias.** *(Left)* Comparison between model trained on balanced cohort vs. automated model segmentation i.e., the full (imbalanced) cohort, showing that a significant fairness gap remains even after balancing the training cohort. *(Right)* OLS regression visualizing the significant positive correlation between age and Dice score.

**Ethnic Disparities and the Masking Effect of Data Aggregation:** A global analysis across the aggregated dataset presents a misleading picture of ethnic fairness. For the Dice score, initial analysis suggests minimal disparity, with a non-significant Kruskal-Wallis test ($H = 5.09$, $p = 0.166$) and a near-equitable DIR of 0.89. Conversely, the HD95 metric indicates a significant disparity against the Asian subgroup ($p = 0.0046$), with a more critical DIR of 0.52. This inconsistency highlights the unreliability of aggregated analysis.

The true extent of bias is only revealed when disaggregating by data source. ANOVA test (`Performance ~ Ethnicity + DataSource`), confirms that the data source is a highly significant variable for both Dice ($F = 11.78$, $p = 1.2 \times 10^{-7}$) and HD95 ($F = 9.10$, $p = 6.0 \times 10^{-6}$). This demonstrates that institutional or cohort-specific factors are major confounders. For example, while the global DPD in Dice scores was only 3.0%, it amplified to 10.0% within the ISPY2 cohort. This disparity, entirely masked by pooling data, reveals that certain ethnic groups face substantial performance degradation in specific clinical contexts.

Additionally, we also find that the interpretation of ethnic bias is dependent on the evaluation metric. For the Dice score, a measure of volumetric overlap, the disparity proved to be an intrinsic bias, as adjusting for the data source only reduced its effect size by 6.2%. In contrast, for the HD95 score, a measure of boundary accuracy, the bias was largely a result of source confounding; adjusting for the data source reduced its effect size by a substantial 64.0%. This divergence suggests the model produces different *types* of segmentation errors for certain ethnic groups.

**Outlook:** We present a comprehensive fairness audit of a breast tumor segmentation model using the multi-center MAMA-MIA dataset, revealing significant age and ethnic disparities. Our analysis identified a persistent intrinsic bias against younger patients that survives balanced training, and severe, site-specific ethnic biases that are masked by multi-center data aggregation. This audit establishes a foundation for investigating the causal mechanisms underlying these biases. Future work will therefore focus on these origins through controlled training experiments and a systematic examination of annotation quality for evidence of label bias, with the ultimate goal of developing targeted mitigation strategies to ensure equitable model performance.

## Potential Negative Societal Impacts

The primary motivation for this work is to mitigate the negative societal impact of biased AI systems in healthcare. Our goal is to promote equity by identifying performance disparities so they can be addressed before deployment. However, we recognize that this research, like any fairness audit, could have unintended negative consequences.

The most direct negative impact stems from the subject of our study itself. If the biases we identify in the segmentation model are not rectified, its deployment in a clinical setting would perpetuate and potentially amplify existing health inequities. Younger patients and certain ethnic minorities would receive a lower quality of diagnostic support, which could lead to delayed diagnoses, suboptimal treatment planning, and ultimately, worse health outcomes. Our work seeks to prevent this exact scenario. Furthermore, there is a risk that our findings could be misinterpreted. A superficial reading might lead to the oversimplified conclusion that "all AI is biased," fostering general distrust in valuable clinical tools.

Despite these risks, we firmly believe that the benefit of transparently reporting these biases far outweighs the potential for misuse. The greatest harm comes from allowing such disparities to remain hidden, where they can silently influence patient care. By bringing these issues to light, we intend to spur corrective action and encourage the development of more robust and equitable medical AI systems.

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
