# OpenReview forum: "Who Does Your Algorithm Fail? Investigating Age and Ethnic Bias in the MAMA-MIA Dataset"
_EurIPS.cc/2025/Workshop/MedEurIPS — EurIPS 2025 Workshop MedEurIPS Submission_

### Official Review · Reviewer_dBjZ · 2025-10-31
**Comprehensive fairness analysis regarding Age and Ethnic Bias in the MAMA-MIA Dataset**

**Rating:** 9
**Confidence:** 4

**Review:**

This work presents the first fairness evaluation for breast cancer tumor segmentation using the MAMA-MIA dataset. The authors evaluate automated segmentation quality across age, ethnicity, and data source, revealing an intrinsic age-related bias against younger patients and ethnic bias.

### Pros

1. It presents the first comprehensive fairness audit of the MAMA-MIA dataset, a large-scale, multi-center breast cancer resource.
2. The analysis across different age groups reveals an intrinsic age-related bias against younger patients that is not explained by data imbalance alone.
3. The paper includes comprehensive statistical analysis to assess performance differences across demographic subgroups, revealing a "masking effect" where aggregating data from multiple sources introduces significant site-specific ethnic biases.

### Cons
1. As suggested by the authors, besides the age group imbalance, there is severe imbalance in ethnicity as well in this dataset. Is it possible that the age bias is influenced by the imbalance of ethnicity as well? For example, if there are more Caucasian patients in the younger group and fewer in the older group, this could confound the age effect. If this is the case, a dual controlled experiment (balancing both age and ethnicity) could answer the question. If not, it would be beneficial to include statistics on the ethnicity distribution within each age group to rule out this potential confound.

2. The paper shows that the interpretation of ethnic bias can change depending on the evaluation metric used (e.g., Dice score vs. HD95) and whether the data is aggregated or not. Does this indicate a limitation in the evaluation metrics themselves? A discussion on the limitations of current metrics and possible alternative evaluation approaches would strengthen the paper.

---

### Official Review · Reviewer_5By3 · 2025-11-03
**review comments.**

**Rating:** 6
**Confidence:** 4

**Review:**

This paper presents a fairness audit studying the intersection of multiple sensitive attributes in the MAMA-MIA breast tumor segmentation dataset.

Strength:
-The authors identify a statistically significant intrinsic bias against younger patients (Age <40), which persists even after controlling for data source and sample imbalance, suggesting a link to complex physiological factors (e.g., breast density).

Suggestion:
It would be interesting by explicitly linking the intrinsic age bias to specific model failure modes (e.g., poorer boundary delineation on dense tissue) to motivate targeted mitigation strategies in future work.

---

### Decision · Program_Chairs · 2025-11-03

**Decision:**

Accept (Oral)

**Comment:**

Both reviewers praise the paper for providing a comprehensive fairness audit of the MAMA-MIA dataset and for uncovering intrinsic age- and ethnicity-related biases in breast cancer segmentation.